

# Errors induced by different approximations in handling horizontal atmospheric inhomogeneities in MIPAS/ENVISAT retrievals

Elisa Castelli[1], Marco Ridolfi[2,6], Massimo Carlotti[3], Björn-Martin Sinnhuber[4], Oliver Kirner[5], and Bianca Maria Dinelli[1]

[1]Istituto di Scienza dell'Atmosfera e del Clima, ISAC-CNR, Bologna, Italy
[2]Dipartimento di Fisica e Astronomia, Universita' di Bologna, Italy
[3]Dipartimento di Chimica Industriale "Toso Montanari", Universita' di Bologna, Italy
[4]Karlsruhe Institute of Technology, IMK-ASF, Karlsruhe, Germany
[5]Karlsruhe Institute of Technology, Steinbuch Center for Computing, Karlsruhe, Germany
[6]Istituto di Fisica Applicata "Nello Carrara", IFAC-CNR, Firenze, Italy

*Correspondence to:* E. Castelli
(e.castelli@isac.cnr.it)

**Abstract.** MIPAS (Michelson Interferometer for Passive Atmospheric Sounding) is a mid-infrared limb emission sounder that operated on board the polar satellite ENVISAT from 2002 to 2012. The retrieval algorithm used by the European Space Agency to process MIPAS measurements exploits the assumption that the atmosphere is horizontally homogeneous. However, previous studies highlighted that this assumption causes significant errors on the retrieved profiles of some MIPAS target species.

5   In this paper we quantify the errors induced by this assumption and evaluate the performances of three different algorithms that can be used to mitigate the problem. We generate synthetic observations with a high spatial resolution atmospheric model and carry out the retrievals with four alternative methods. The first assumes horizontal homogeneity (1D retrieval), the second includes a model of the horizontal gradient of atmospheric temperature (1D plus temperature gradient retrieval), the third accounts for an horizontal gradient of temperature and composition (1D plus temperature and composition gradient retrieval), while the fourth is the full two-dimensional (2D) inversion approach.

Our results highlight that the 1D retrieval implies errors that are significant for averages of profiles. Furthermore, for some targets (e.g. T, $CH_4$ and $N_2O$ below 10 hPa) the error induced by the 1D approximation becomes visible also in the individual retrieved profiles. The inclusion of any kind of horizontal variability model improves all the targets with respect to the horizontal homogeneity assumption. For temperature, $HNO_3$ and CFC-11 the inclusion of an horizontal temperature gradient leads to

15   a significant reduction of the error. For other targets as $H_2O$, $O_3$, $N_2O$, $CH_4$, the improvements due to the inclusion of an horizontal temperature gradient are minor. In these cases, the inclusion of a gradient in the target volume mixing ratio leads to significant improvements. Among all the methods tested in this work, the 2D approach, as expected, implies the smallest errors for almost all the target parameters. This residual error of the 2D approach is the smoothing caused by the retrieval grid, that is coarser than that of the atmospheric model.



# 1 Introduction

Satellite limb scanning spectrometers have been widely used to measure atmospheric composition and its evolution with time. In many cases the atmospheric composition is obtained from these measurements with retrieval schemes that assume a horizontally homogeneous atmosphere (the so called one-dimensional, or 1D retrieval approach). In most of the stratosphere
this assumption does not produce huge systematic errors. However, in the case of strong horizontal variability, retrieved profiles may be affected by significant error.

In order to quantify this error, we exploit the measurements of the MIPAS (Michelson Interferometer for Passive Atmospheric Sounding) instrument, that operated onboard the ENVISAT satellite from March 2002 to April 2012. The instrument observed the atmospheric mid-infrared emission spectrum using the limb-scanning observation technique (Fischer et al., 2008).
In this spectral region several minor atmospheric constituents are active, and from the inversion of the spectrum it is possible to determine their volume mixing ratio (VMR) vertical profile in the height range from 6 to 70 km. Due to the polar orbit of ENVISAT, MIPAS measured during day-time in the descending (DX) part of the orbit and during night-time in the ascending (AX) part. Comparing MIPAS/ENVISAT CFC-11 profiles with ACE (Atmospheric Chemistry Experiment) matching measurements, Höpfner et al. (2007) discovered unrealistic differences between MIPAS night-side and day-side CFC-11 VMRs. These
discrepancies were thoroughly investigated by Kiefer et al. (2010) and were attributed to the unmodelled horizontal variability of the atmosphere in the ESA Level 2 processor. Similar horizontal gradients of temperature or composition are sounded by the instrument line of sight with opposite sign in the ascending and the descending parts of the orbit. This sign difference causes opposite systematic errors in the profiles retrieved from the measurements acquired in the ascending and descending parts of the orbits.
In order to account for the horizontal inhomogeneities of the atmosphere, tomographic inversion codes with the capability of a full two-dimensional (2D) model were developed for MIPAS (e.g. GMTR (Carlotti et al., 2006), MORSE (Dudhia et al., 2005), RET2D (von Clarmann et al., 2003) and the 2D-option of the RCP (Steck et al., 2005)), for SCIAMACHY (Scanning Imaging Absorption Spectrometer for Atmospheric Chartography) (Pukite et al., 2008) and for MLS (Microwave Limb Sounder) (Livesey et al., 2006). Given the complexity of the 2D retrieval models, von Clarmann et al. (2009) and Kiefer et al.
(2010) evaluated the representation of the horizontal variability of the atmosphere with a temperature (T) horizontal gradient model in the MIPAS 1D retrieval codes. However, to date, the impact of neglecting the horizontal variability of the atmosphere and the relative improvements that can be achieved including a horizontal gradient model are still to be quantified. In this study we quantify the error due to the 1D assumption on MIPAS retrievals on the basis of synthetic observations produced with a highly resolved atmosphere (1.4° in latitude) derived from the chemistry climate model EMAC (ECHAM/MESSy Atmospheric Chemistry, Jöckel et al. (2006)) and the 2D FM (Forward Model internal to the GMTR system). We retrieve the
atmospheric state from these synthetic observations using, alternatively, the standard GMTR algorithm and three ad-hoc modified versions of the same code. The first modified version accounts for the horizontal variability by including user-supplied a-priori horizontal gradients of temperature, the second includes user-supplied gradients of atmospheric composition, while





the third modified version emulates the 1D retrieval algorithm. The differences between the retrieved and the true atmospheric state profiles provide an estimate of the error caused by the retrieval approach.

The paper is structured as follows: in Sect. 2 we describe the method used to evaluate the size of the errors implied by the different retrieval approaches. In Sect. 3 we discuss the results of our tests. Finally, the conclusions are given in Sect. 4.

## 2   Method

When analysing real measurements the true atmospheric state is not known. For an accurate assessment of the systematic retrieval errors due to different approaches used to account for the horizontal variability, we therefore use synthetic observations based on data from a high resolution simulation of EMAC and on a 2D FM which is as accurate as possible. We then apply different retrieval algorithms to these observations and evaluate the errors on the basis of the differences between retrieved and reference target parameters.

### 2.1   Atmospheric model

The reference atmosphere used to produce the synthetic spectra was extracted from a high resolution (T85 truncation, corresponding to a horizontal resolution of 1.4°x1.4°) simulation of the chemistry climate model EMAC (version 1.10; Jöckel et al. (2006)) for the year 2011. In the EMAC simulation, a Newtonian relaxation technique towards meteorological analyses for the prognostic variables temperature, vorticity, divergence and the surface pressure was implemented to reproduce realistic synoptic conditions. We applied this nudging technique using the ERA-Interim reanalysis (Dee et al., 2011) from the European Centre for Medium-range Weather Forecasts (ECMWF). In this EMAC simulation a comprehensive treatment of tropospheric and stratospheric chemistry was included. Model output includes temperature, pressure, geopotential height, $H_2O$, $O_3$, $HNO_3$, $N_2O$, $CH_4$, $NO_2$, CFC-11, CFC-12, $N_2O_5$, $ClONO_2$ and $CCl_4$ on 39 model levels. The geopotential height at the surface was also given and used for the conversion to geometric height. To investigate possible seasonal variability of the error due to horizontal variability, we selected the atmospheres for the days corresponding to the solstices and equinoxes of the year 2011. For each of these days, EMAC profiles corresponding to the geolocation of real MIPAS limb scans were extracted. In particular, we selected the geolocation of MIPAS measurements in the orbits 47349 and 47350 for March 21, 48671 and 48672 for June 21, 49964 and 49965 for September 21, 51301 and 51302 for December 21. Figure 1 shows an example of temperature, $H_2O$, $O_3$, $HNO_3$, $CH_4$ EMAC distribution for 21 December 2011 along MIPAS orbit 51301. The atmospheric horizontal variability is clearly visible in Fig. 1.

The results of Kiefer et al. (2010) are based on the assumption that the average atmosphere in a given latitude band should not depend on longitude, therefore with a perfect retrieval scheme the zonal averages of retrieved parameters should be equal when computed from measurements in the ascending or descending parts of the orbit. In the case of retrieval schemes that do not consider properly the horizontal variability, a systematic difference between the averaged ascending and the descending measurements will appear. We call this difference "AX-DX difference". In order to more clearly simulate this issue, we modified the above mentioned reference atmospheres to make them identical in the ascending and descending parts of each orbit.



If the error due to the measurement noise is sufficiently small, observed AX-DX differences in the retrieved data can therefore be exclusively attributed to the treatment of the horizontal variability in the retrieval.

## 2.2 Synthetic observations

The synthetic observations used in our analysis are produced with the 2D FM internal to the GMTR code. In this FM the discretization of the atmosphere is sufficiently fine to accurately model the small-scale structures of the reference atmosphere (Fig. 1) described in Sect. 2.1.

In order to focus our analysis on the error caused solely by the approximations in modelling the horizontal variability of temperature and the target gas, for the generation of synthetic observations we modified the reference atmosphere as follows: The observations used for the retrieval of the VMR of a given gas were generated assuming for both temperature and target gas the 2D distributions of the reference atmospheric model of Sect. 2.1. The vertical distributions of pressure and the other interfering, non-target gases were set constant in the horizontal domain, equal to the profiles of the reference model at ≈45° N latitude. Similarly, the observations used for the joint retrieval of pressure and temperature (pT retrieval) are generated assuming the 2D temperature distribution of the reference atmospheric model of Sect. 2.1, while the profiles of pressure and the VMR of all atmospheric species are forced to be constant in the horizontal domain, equal to the profiles of the reference model at ≈45° N latitude.

From previous studies (mentioned in Sect. 1) we know that the size of the error induced by the 1D assumption in MIPAS retrievals is comparable to that of the error due to measurement noise. Therefore, to better highlight the error induced by the horizontal variability model we avoided the masking effect of the noise error by adding to the synthetic observations a pseudo-random noise with standard deviation much smaller (a factor of 40) than that of real MIPAS measurements.

## 2.3 Retrieval algorithms

We carried out the retrievals on these synthetic observations using four algorithms. All of them were based on the GMTR algorithm of Carlotti et al. (2006).

The four algorithms, however differ in modelling the horizontal variability of the atmosphere: In the 1D algorithm the atmosphere is assumed horizontally homogeneous; in the 1D+grad(T) algorithm the atmosphere is assumed horizontally homogeneous but a prescribed horizontal temperature gradient is included; in the 1D+grad(T,VMR) algorithm the atmosphere is assumed horizontally homogeneous but a prescribed horizontal gradient is included for temperature and the VMR of the target species. Finally, the full 2D atmospheric variability is modelled in the 2D algorithm.

While in the first three retrieval methods each limb scan is processed individually with a Global Fit (Carlotti, 1988) approach, in the 2D retrieval the measurements of a full orbit are simultaneously used for the retrieval of the 2D distribution (Carlotti et al., 2006). Furthermore, while in the second and third retrieval methods the horizontal gradients are externally provided by the user (fixed gradients), in the 2D retrieval they are represented by the 2D retrieved distribution.

The horizontal gradients used in the 1D+grad(T) and 1D+grad(T,VMR) tests are obtained from the 1D retrieved atmosphere. Two different horizontal gradients, altitude dependent, are calculated for the two portions of the line of sight before and beyond





the tangent point of each limb observation. The gradients are used only in a range of ±400 km about each tangent point. The profiles outside this region are set equal to their values at the boundaries of the region itself.

Note that, in the 2D approach, a vertical profile was retrieved at the average position of each measured limb scan. Since the spacing between adjacent limb scans is of the order of 400 km, it follows that the 2D retrievals are affected by the so-called

*smoothing error* due to a retrieval grid step coarser than the step adopted in the reference atmosphere.

The target profiles we retrieve from the synthetic observations are pressure, temperature (pT joint retrieval), the VMR of $H_2O$, $O_3$, $HNO_3$, $N_2O$, $CH_4$ and CFC-11. For the assessment of the error for each target we first retrieve pT and use it in the subsequent retrieval of the VMR of the target gas under consideration. All the retrievals are performed on a fixed altitude grid coinciding with the nominal limb-scan of the MIPAS mission after January 2004 (Raspollini et al., 2013). The altitudes of the

grid are: 72, 66, 62, 58, 54, 50, 46, 43, 40, 37, 34, 31, 29, 27, 25, 23, 21, 19.5, 18, 16.5, 15, 13.5, 12, 10.5, 9, 7.5, 6 km. The retrieval analysis are operated on 3 cm$^{-1}$ spectral intervals (the so called Micro Windows, MWs) containing information on the target parameters (Raspollini et al., 2013) and are the same as those used for MIPAS operational data processing with the ESA Level 2 processor Version 6.

## 2.4 Quantifiers

In order to characterize the performance of the different approaches employed to model horizontal variability, we first group the profiles resulting from synthetic retrievals ($x_{ret}$) in 15-degrees latitude bands, spanning the range from 90° N to 90° S. The latitude bands are numbered with the index $k = 1, ..., 12$. In each latitude band $k$, we then interpolate the retrieved profiles to fixed pressure levels $i = 1, ..., n_p$ and, separately for the profiles belonging to the AX and DX parts of the orbits, we calculate, for each pressure level, the following quantity:

$$diff(i,k) = \frac{\sum_{j=1}^{n_k}(x_{ret}(i,j) - x_{ref}(i,j))}{n_k} \qquad (1)$$

where $j = 1, ..., n_k$ numbers the retrieved ($\mathbf{x}_{ret}(i,j)$) and reference ($\mathbf{x}_{ref}(i,j)$) profiles within each latitude band $k$. The quantity defined in Eq. (1) represents the estimated bias, as a function of pressure and latitude band. The difference between $diff(i,k)$ evaluated for the AX and DX parts of the orbits, mimics the "AX-DX difference" analyzed by Kiefer et al. (2010), i.e. the most obvious artefact introduced by retrieval algorithms assuming horizontal homogeneity of the atmosphere.

In order to evaluate the performance of the horizontal variability models in different measurement conditions, we also group the results of synthetic retrievals according to the following classes: a) from 60° N to 90° N in December and March, and latitude from 60° S to 90° S in September and June (Polar Winter), b) from 60° N to 90° N in September and June, and latitude from 60° S to 90° S in December and March (Polar Summer), c) from 60° N to 30° N and from 60° S to 30° S, all seasons (Mid latitude), d) from 30° N to 30° S, all seasons (Equatorial) and finally e) the Whole Orbit class including all latitudes and

seasons. For each of these classes we evaluate the Root Mean Square Error (RMSE).



## 3 Results and Discussion

In Fig.s 2, 3 and 4 the average AX-DX differences in the 60° S - 45° S latitude band for temperature and different target species are shown. In agreement with Kiefer et al. (2010), the AX-DX differences significantly deviate from zero for all the considered target species from the 1D retrieval. Since the reference atmosphere used to generate the synthetic observations

was symmetric with respect to the South Pole and the measurement noise was very small, the observed differences can only be ascribed to the biases introduced by neglecting the horizontal variability of the atmosphere in 1D retrievals. The same figures show the AX-DX differences obtained with the 1D+grad(T) retrieval, the 1D+grad retrieval, and the 2D retrieval. We see in these figures that the introduction of a model for horizontal temperature gradients already reduces significantly the observed AX-DX differences in temperature, $HNO_3$ and CFC-11. This result is in good agreement with the results of Fig. 17 of Kiefer et

al. (2010) that refers to an average of nine days of MIPAS measurements in January 2003, in the same latitude band. Actually, the AX-DX differences we found in case of both the 1D and the 1D+grad(T) approaches are very similar to those of Kiefer et al. (2010). This similarity holds despite the fact that our test case and the one of Kiefer et al. (2010) differs in several aspects. For example: a) we use simulated instead of real data, b) we consider different years and months, c) we use horizontal gradients obtained from a previous 1D retrieval while Kiefer et al. (2010) uses gradients obtained from ECMWF re-analysis. This result

suggests that the observed AX-DX differences depend mainly on the actual amplitude of the horizontal gradients, which in turn, depends mainly on season.

Figs. 2, 3 and 4 also show that modelling the horizontal VMR gradients causes a further reduction of the AX-DX differences. Finally, 2D retrievals produce AX-DX differences close to zero for most of the considered target species, as was already observed in Kiefer et al. (2010) using real MIPAS measurements.

Figs. 5, 6 and 7 show the RMSE as a function of altitude for temperature, $H_2O$, $O_3$, $HNO_3$, $N_2O$, $CH_4$, and CFC-11 for the different retrieval approaches. In each panel of these figures, the gray dot-dashed line represents the average noise error obtained in the standard Level 2 processing. This error will be referred to as "RND" error and is reported in the plots to evaluate the significance of the error obtained with the different models of horizontal variability. The right panels of these figures refer to the "whole orbit" scenario described in Sect. 2.4, considered to provide an overall picture of the investigated model error.

The averages of the RMSE's over selected pressure ranges for each target are reported in Table 1.

In the case of temperature, the 1D method provides largest RMSE in Polar Winter and Mid Latitudes conditions and the smallest RMSE in the Equatorial scenario (Fig. 5). As expected, this result correlates directly with the amplitude of the horizontal temperature gradients. The RMSE obtained for temperature in 1D retrievals is always larger than the RND error, with the exception of the pressure range from 10 to 70 hPa ($\approx 19-31$ km) in the Equatorial region. Over the whole orbit scenario,

the RMSE averaged over the whole pressure range is about 1.2 K. For 1D $H_2O$ retrieval (Fig. 5) the smallest RMSE is obtained in Polar Summer conditions, and in general the RMSE is well below the RND error. Over the whole orbit scenario, the RMSE averaged over the pressure range 0.1-70 hPa is about 5 % while it is 16 % in the 70-200 hPa range(see Table 1). In the case of $O_3$(Fig. 6), the worst results are obtained with 1D retrievals for Polar Winter and Mid Latitudes conditions (in agreement with the latitudinal behaviour of the horizontal temperature and ozone gradients) with an average RMSE over the whole orbit





scenario of the same order as the RND error. RMSE calculations for 1D $HNO_3$ retrievals (Fig. 6) show that the largest RMSEs are obtained for Polar Winter and Mid Latitudes conditions. In these cases the RMSE is larger than the RND error. A similar behaviour is observed for $N_2O$ and $CH_4$ (Fig. 7) with an average RMSE of 5 %. Finally, for CFC-11 we get an average RMSE of 7 % in the 25-100hPa region and 3 % in the 100-300hPa region when using the 1D approach.

These results show that the introduction of all of the models for horizontal variability produce improvements for all the targets (Table 1). The temperature RMSEs are reduced to less than 1 K when horizontal gradients are modelled and to less than 0.5 K in the case of 2D retrievals. $H_2O$ RMSEs can be reduced down to 3 % in the 0.1-70 hPa region and to 10 % in the 70-200 hPa region by modelling gradients. The error on $O_3$ reduces to a few percent below 40 hPa and above 0.2 hPa (see Fig. 6). In the case of $HNO_3$, both modelling of horizontal gradients and the 2D approach mitigate the effect of horizontal variability

(only few percent differences remain) over the whole altitude range. In the case of $N_2O$ and $CH_4$, modelling the T and VMR horizontal gradients and the 2D approach greatly reduces the error, in particular in the altitude range of their peak VMRs. The CFC-11 1D retrieval is heavily influenced by the horizontal variability with a large (around 15 %) error above 100 hPa. This error can be greatly reduced when atmospheric variability is taken into account (Fig. 4 and Fig. 7).

    The modelling of a temperature horizontal gradient produces a significant reduction of the error in temperature, $HNO_3$,

CFC-11. $H_2O$ retrievals also improve partially at high altitudes. The reduction of error is marginal for the other VMR retrievals. Modelling also the horizontal VMR gradient produces a further reduction of the RMSE. The largest improvements are obtained in $O_3$, $CH_4$ and $N_2O$ retrievals, while $H_2O$ at high altitudes and CFC-11 benefit very little from this additional sophistication.

    The RMSE's obtained with the 2D approach, reported in the last column of Table 1, are generally smaller than those obtained with any of the simpler horizontal gradient models. Improvements given by the use of different approaches to model horizontal

gradients with respect to 1D case are summarised in Table 2.

## 4   Conclusions

In this work we quantify the error induced by neglecting the horizontal variability of the atmosphere in MIPAS retrievals, and characterize possible alternative retrieval approaches that could help to reduce this error. Our study is based on synthetic limb observations generated with an accurate 2D forward model that assumes a known atmospheric state taken from a high resolution model (EMAC). We evaluate the relative performance of some different retrieval approaches: the simple 1D model,

the model of horizontal temperature gradients, the model of both temperature and a VMR horizontal gradients, the full 2D model.

    The results show that neglecting the horizontal atmospheric variability can produce average errors of $\approx$1.2 K for temperature (maximum around 2 K), 5-16% for $H_2O$, 3-11% for $O_3$, 6-15% for $HNO_3$, 6% for $N_2O$, 5% for $CH_4$, 7-3% for CFC-11

depending on the considered pressure range. In case of T, $CH_4$ and $N_2O$ retrievals below 10 hPa the error induced by the 1D approximation becomes significant compared to noise error also in the individual retrieved profiles. For the other target parameters this effect is expected to be important only for averages of profiles.





Modelling a temperature horizontal gradient improves temperature (error reduced to 0.6 K), $HNO_3$ and CFC-11 retrievals (as already reported in (Kiefer et al., 2010)). The effect on the retrieval of the other considered molecules is minor. As already mentioned in Carlotti et al. (2013), this is due to the fact that the 1D retrieved temperature profiles, even if differing from the real temperature, represent "effective" profiles that partially compensate for the horizontal inhomogeneity error. Thus,

modelling the horizontal temperature gradient improves the accuracy of the temperatures but only marginally the quality of some retrieved VMRs. The amplitude of this compensation effect depends on MWs used.

Modelling both temperature and VMR gradients reduces the error to 3-10% for $H_2O$, 2-9% for $O_3$, 4-9% for $HNO_3$, 4% for $N_2O$, 3% for $CH_4$ while no improvement is obtained for CFC-11. The improvements obtained for $O_3$, $CH_4$ and $N_2O$ are particularly important, also because they are concentrated in the altitude region where the VMRs have their maximum values.

The horizontal temperature and VMR gradient values used in our tests are derived from the corresponding atmospheric fields retrieved with the 1D assumption. We consider these gradients relatively accurate. We verified, however, that using less accurate gradients, such as those that can be inferred e.g. from ECMWF analyses, usually produces a less significant reduction of the retrieval error.

The 2D retrieval approach produces the smallest error in modelling the horizontal variability of the atmosphere. We note

that the real benefits of the 2D approach are even more evident when looking at the vertical distribution of the errors: the 2D results are always performing better than the other approaches, especially in the altitude regions and latitudinal bands where the horizontal variability is largest. The remaining error is due to the horizontal smoothing intrinsic to the measuring system. With the adopted atmospheric model, this smoothing error is of the order of: 0.5 K for temperature, 2-8% for $H_2O$, 2-7% for $O_3$, 4-10% for $HNO_3$, 4% for $N_2O$, 3-1% for $CH_4$, 3-2% for CFC-11. In the 2D approach, all the measurements of a full orbit

are simultaneously inverted to infer the full 2D distribution of the target parameter. Therefore the 2D approach does not need any externally supplied horizontal gradients and, for this reason, it is not prone to the systematic error that may be caused by the use of inaccurate external information.

*Acknowledgements.* This work was performed under ESA-ESRIN Contract no. 21719/08/I-OL. The authors gratefully acknowledge Michael Kiefer (KIT) for useful comments, Richard Siddans (RAL) for proofreading the manuscript and ECMWF for access to data.



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



**Table 1.** Errors due to different treatment of horizontal inhomogeneities (1D, 1D+grad(T), 1D+grad(T,VMR), 2D) for each target.

|  | 1D | 1D+grad(T) | 1D+grad(T,VMR) | 2D |
|---|---|---|---|---|
| Temperature [K] | 1.2 | 0.6 | – | 0.4 |
| $H_2O$ (0.1-70hPa) [%] | 5. | 3. | 3. | 2. |
| $H_2O$ (70-200hPa) [%] | 16. | 12. | 10. | 8. |
| $O_3$ (0.1-40hPa) [%] | 3. | 2. | 2. | 2. |
| $O_3$ (40-300hPa) [%] | 12. | 11. | 10. | 7. |
| $HNO_3$ (3-40hPa) [%] | 6. | 4. | 4. | 4. |
| $HNO_3$ (40-200hPa) [%] | 15. | 12. | 9. | 10. |
| $N_2O$[%] | 6. | 5. | 4. | 4. |
| $CH_4$ (0.1-10hPa) [%] | 5. | 4. | 3. | 3. |
| $CH_4$ (10-300hPa) [%] | 5. | 4. | 3. | 1. |
| CFC-11 (25-100hPa) [%] | 7. | 3. | 3. | 3. |
| CFC-11 (100-300hPa) [%] | 3. | 2. | 3. | 2. |

**Table 2.** RMSE improvements due to different treatment of horizontal inhomogeneities (1D+grad(T), 1D+grad(T,VMR), 2D) for each target with respect to 1D.

|  | 1D+grad(T) vs 1D | 1D+grad(T,VMR) vs 1D | 2D vs 1D |
|---|---|---|---|
| Temperature [%] | 49 | – | 63 |
| $H_2O$ (0.1-70hPa) [%] | 38 | 37 | 65 |
| $H_2O$ (70-200hPa) [%] | 22 | 37 | 49 |
| $O_3$ (0.1-40hPa) [%] | 27 | 33 | 35 |
| $O_3$ (40-300hPa) [%] | 9 | 19 | 44 |
| $HNO_3$ (3-40hPa) [%] | 36 | 37 | 36 |
| $HNO_3$ (40-200hPa) [%] | 19 | 37 | 33 |
| $N_2O$[%] | 20 | 34 | 37 |
| $CH_4$ (0.1-10hPa) [%] | 13 | 24 | 24 |
| $CH_4$ (10-300hPa) [%] | 25 | 38 | 71 |
| CFC-11 (25-100hPa) [%] | 54 | 57 | 57 |
| CFC-11 (100-300hPa) [%] | 35 | 19 | 49 |





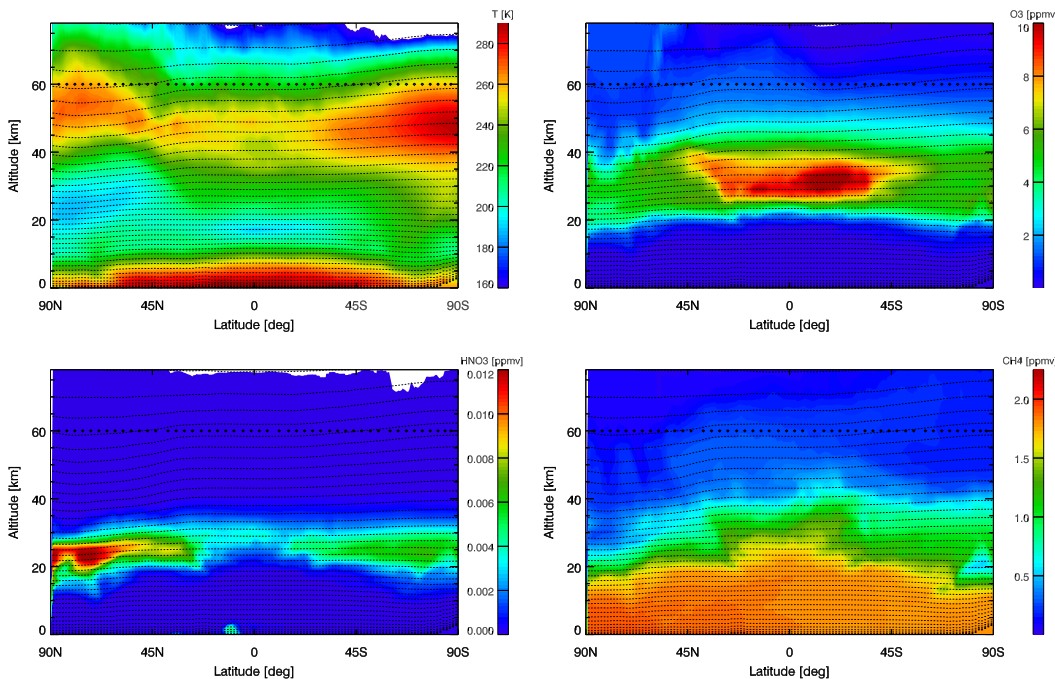

**Figure 1.** Temperature, $H_2O$, $O_3$, $HNO_3$ and $CH_4$ distributions from EMAC model simulations for the December 21, 2011 as a function of altitude and latitude (90° N-90° S) along MIPAS orbit 51301. The small black dots represent EMAC altitude-latitude grid, while black diamonds represent the latitudinal position of MIPAS scans in this orbit.



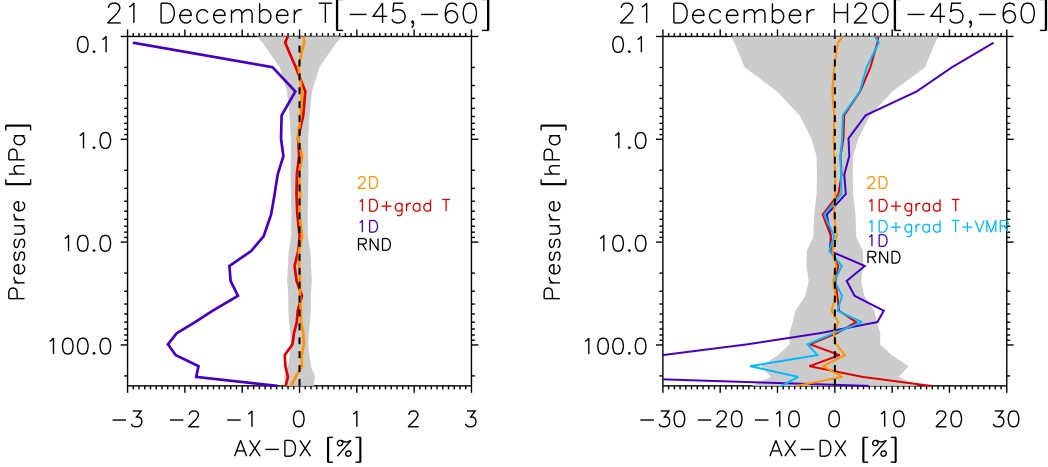

**Figure 2.** AX-DX differences for December 21, 2011 in the 45° S-60° S latitude band for T and H$_2$O. The line colours show: in blue the 1D retrievals, in red the 1D retrieval with gradients, in cyan the 1D retrieval with gradients of T and target species, and in orange the 2D retrievals. The grey shadow represents the average ORM random error (RND).

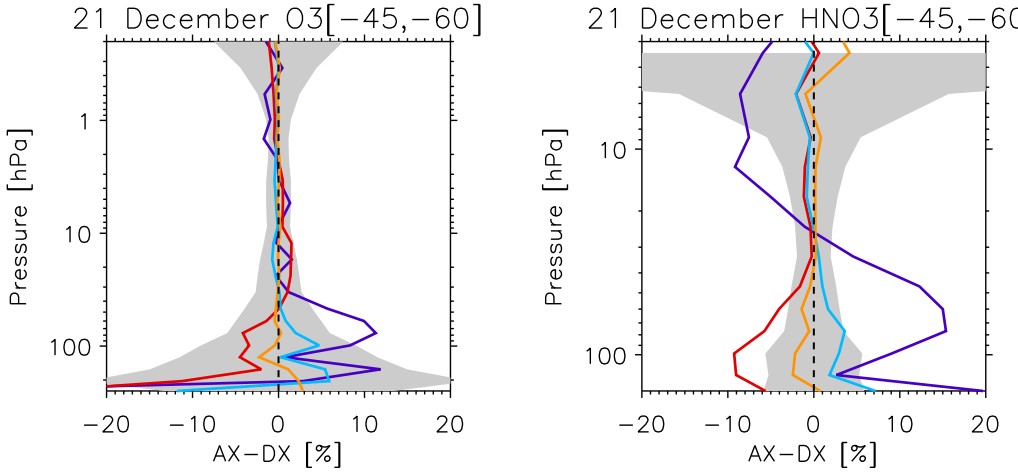

**Figure 3.** AX-DX differences for December 21, 2011 in the 45° S-60° S latitude band for O$_3$ and HNO$_3$. The colour code is the same as in Fig. 2.



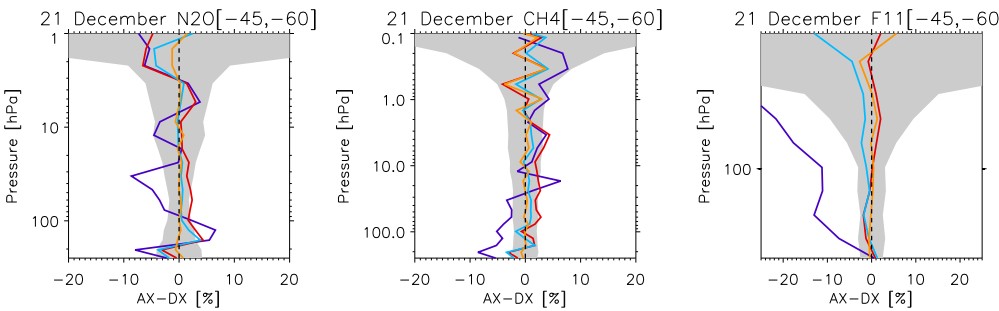

**Figure 4.** AX-DX differences on 21 December 2011 in the 45° S-60° S latitude band for N$_2$O, CH$_4$, and CFC-11. The colour code is the same as in Fig. 2.





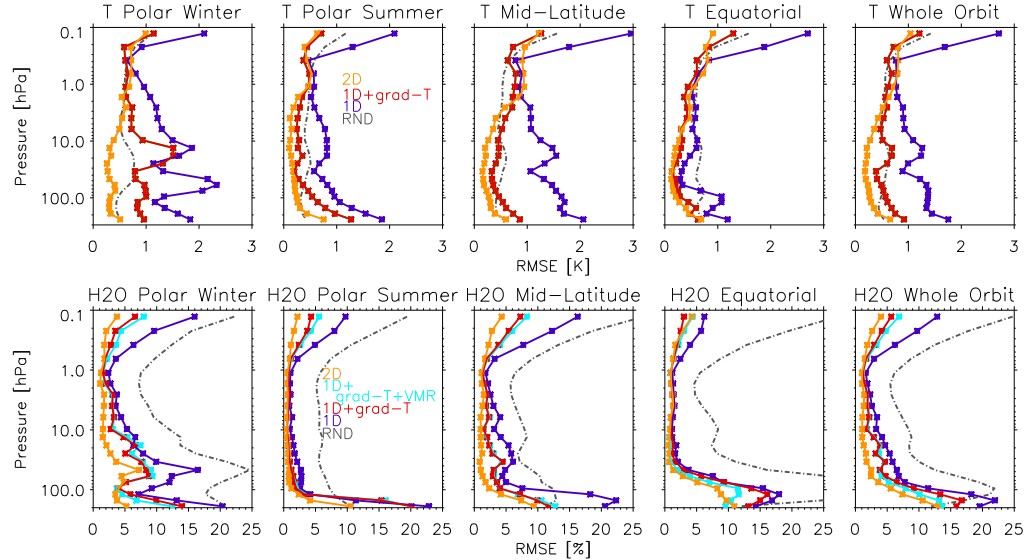

**Figure 5.** RMSE calculation for temperature (upper panels) and $H_2O$ (lower panels) in Polar Winter, Polar Summer, Mid-Latitudes, Equatorial and whole orbit scenario. Blue for 1D, red for 1D+grad(T), cyan for 1D+grad(T,VMR), and orange for 2D. Grey line represents the average ORM random error (RND).





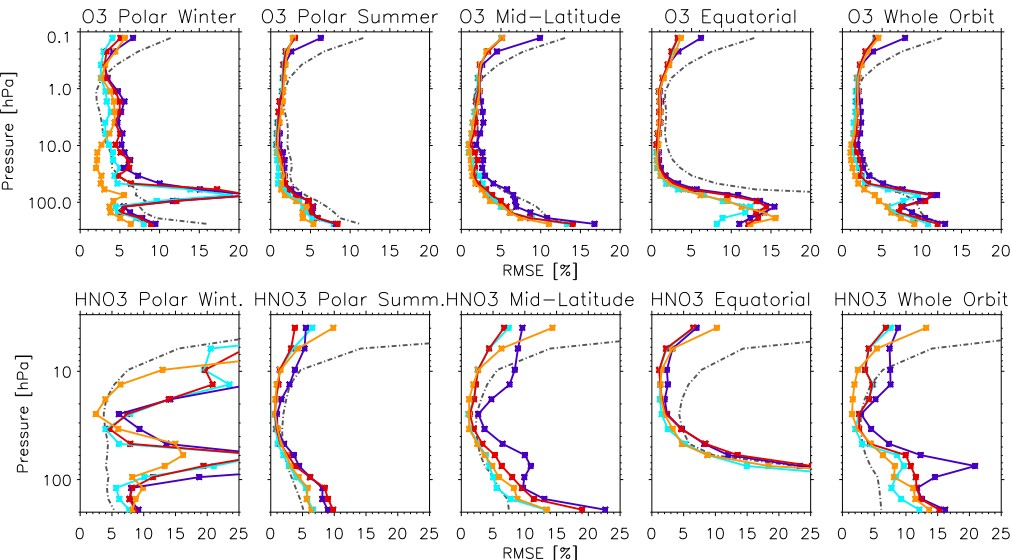

**Figure 6.** RMSE calculation for O$_3$ (upper panels) and HNO$_3$ (lower panels) in Polar Winter, Polar Summer, Mid-Latitudes, Equatorial and whole orbit scenario. The colour code is the same as in Fig. 5.



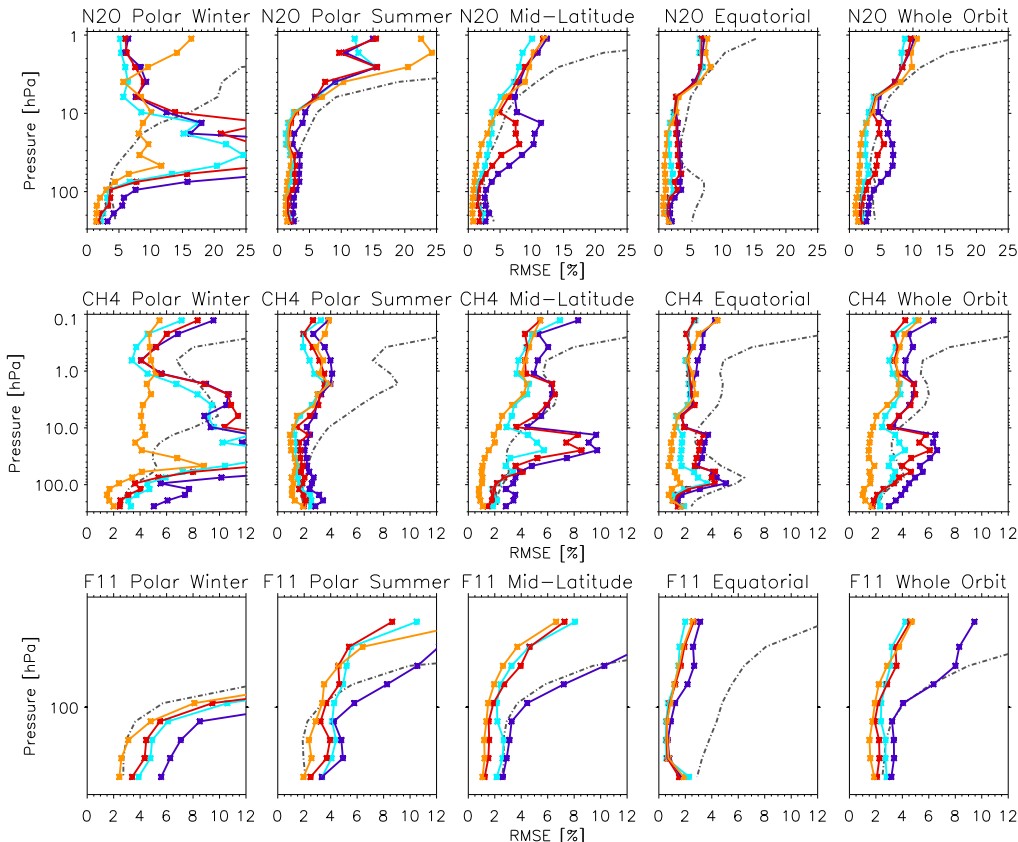

**Figure 7.** From Top to Bottom: RMSE calculation for $N_2O$, $CH_4$, CFC-11 in Polar Winter, Polar Summer, Mid-Latitudes, Equatorial and whole orbit scenario. The colour code is the same as in Fig. 5.