# Peer review of "Errors induced by different approximations in handling horizontal atmospheric inhomogeneities in MIPAS/ENVISAT retrievals"

_Atmospheric Measurement Techniques, 2016_

## Referee Comment (RC1) · Anonymous Referee #1 · 24 May 2016

The manuscript "Errors induced by different approximations in handling horizontal atmospheric inhomogeneities in MIPAS/ENVISAT retrievals" by E. Castelli et al. evaluates several approaches to account for the horizontal atmospheric inhomogeneity. Many limb algorithms assume that the atmospheric parameters and gas concentrations do not vary horizontally along the line of sight. However, this assumption can produce significant errors in the retrieved profiles, especially over latitudes and seasons where large horizontal gradients in atmospheric parameters are observed. In this study authors test several approaches to account for the horizontal inhomogeneity and provide error estimates for each method. This study fits to the scope of the problems considered in AMT. The paper is well organized and written for most of the part.

[Figure]

The methods and results are fully explained. The manuscript is recommended for the publication after some minor corrections.

Specific comments: Throughout the manuscript authors use many acronyms, but not all of them are explained. Please, spell out all acronyms where they are used for the first time in the text. page 2, l.21-23, Please, spell out acronyms 'GMTR', 'MORSE', 'RET2D' and 'RCP'. You use 'GMTR' several times later in the text. Figure 2, Please, spell out 'ORM'.

Page 4, Section 2.2, lines 9-15: I found that this part of the section is not clearly written. Did you set the pressure profile to be the same for all latitudes in order to reduce a natural noise and to better isolate an error due to the 1D assumption? Please, explain that in the text.

Page 7, lines 5-14. I feel this part needs some revisions. For example, the text says "H2O RMSEs can be reduced down to 3 % in the 0.1-70 hPa region and to 10 % in the 70-200 hPa region by modeling gradients". By modeling gradients in T or T+VMR? "The error on O3 reduces to a few percent below 40 hPa and above 0.2 hPa (see Fig. 6)." Why did the O3 error reduce? Did it reduce as a result of modeling temperature gradients or VMR gradients? "This error can be greatly reduced when atmospheric variability is taken into account (Fig. 4 and Fig. 7)". Did you mean the 1D + gradients approach here or the full 2D retrieval?

Figure1. Please, add a title on each panel of Fig. 1, for example "Temperature". You have labels on color scales, but they are too small to see.

Figures 3, 4, 6 and 7: Please, add labels for each color lines. You have these labels on Figs.2 and 5.

Figure 5. I would recommend to spell out 'RMSE' here for readers convenience, even though you did that in the text.

---

## Referee Comment (RC2) · Anonymous Referee #2 · 27 May 2016

Review of Errors induced by different approximations in handling horizontal atmospheric inhomogeneities in MIPAS/ENVISAT retrievals

by Castelli and collagues.

This is a nice paper that clearly describes a solid study of the impact of horizontal gradients on various approaches to retrievals from the MIPAS instrument. In principal I am happy to see this manuscript proceed to AMT, however, I only have one concern that I'd like to understand beforehand (plus some minor suggestions/comments for the authors to consider). The standard of English is excellent and the figures are very clear.

[Figure]

My main concern surrounds the discussion on page 4, lines 10-15. As I understand it, the authors have taken model temperature and pressure profiles on a fixed height grid, and forced the pressures to be horizontally homogeneous while retaining the model horizontal temperature variability. If the altitude grid indeed remains fixed, then surely the model atmosphere that results is not in hydrostatic balance. As such, this becomes an unphysical simulation from which it is arguably hard to draw meaningful "real world" insights. Furthermore, are the lines being observed not subject to significant pressure broadening, making pressure the dominant contributor to the radiance signals? As such, I would have thought that horizontal gradients in pressure are arguably the most important thing to study the impact of (though one would probably ultimately quantify it in terms of impact to the temperature/composition profiles as interpolated to a fixed pressure grid, being the product most widely used in the community). While I understand the authors desire to "focus the analysis on the error caused solely by the approximations in modeling the horizontal variability of temperature and target gas", surely, if the pressure gradients are indeed the dominant term, they should have been included in the analysis.

I'd like the authors to address this point, and consider revising their approach, or at least undertaking a separate quantification of the impact of pressure gradients (but again, the unphysical nature of their atmosphere would limit the usefulness of the result). Perhaps I have misunderstood the description in the manuscript, in which case, greater clarity is required.

==== Minor comments

—- Page 3

Lines 27/28: I don't understand this sentence. By "average atmosphere in a given latitude band" do you mean a zonal mean, i.e., an average over all longitudes? If so, hasn't longitude, by definition, been ruled out. Do you mean averages over a longitude range should not depend on the choice of range (clearly not an appropriate assumption

geophysically). Please clarify.

—- Page 4

Line 5: Please be more quantitative; what is the spatial resolution used in the FM?

Lines 10-15: Please see discussion above.

Lines 17-19: First, please explain why you needed to add any noise at all? Why not simply do a noise free simulation. Also, please state whether, in addition to adding 1/40th of the expected noise, you also quote that 1/40th value as the radiance precision in the retrieval calculation ($S\_y$ in the Rodgers formulation), or does the retrieval still believe that the noise is at the 100% level?

—- Page 5

Line 20 (and 22). Your "diff" has been typeset in math mode, you'll want to typeset it in text mode (e.g., \text{diff}, using the amsmath.sty package).

—- Page 6

Line 26: I think this would be better "Polar and mid-latitude winter conditions" if that's what you mean. As it is it could mean "Polar winter" and "mid-latitude all seasons".

—- Page 7

Line 2: As for page 6, line 26. Possibly elsewhere that I didn't catch also.

—- Figures

Figures 2 and on: I would much prefer to see temperature errors quantified in K than in %. K are much more accessible to the general reader.

---

## Referee Comment (RC3) · Anonymous Referee #3 · 6 Jun 2016

Review of "Errors induced by different approximations in handling atmospheric inhomogeneities in MIPAS/ENVISAT retrievals" by Castelli et al.

This paper provides simulated results that describe the need for a two-dimensional retrieval approach to infer vertical information from MIPAS limb emission measurements. The authors simulate measurements and perform a variety of different retrievals in an attempt to quantify the errors induced by an assumption of horizontal homogeneity within the retrieval scheme. The simulations are by no means comprehensive but the reader is left with the impression that any retrieval scheme that uses two-dimensional information is less likely to produce artefacts related to atmospheric structure along the satellite track. This is of course by no means a surprising result. Although this paper

provides no real information it is another work that supports the hypotheses that in order to accurately retrieve atmospheric composition information from limb measurements, a set of dense observations and a multi-dimensional retrieval are required.

Major Comments:

I found the paper to be well written and the information that was presented was done so in an organized and easily understood manner.

It is very unclear to me what I am supposed to have learned from reading this paper. The limited number of simulations performed do not allow me to quantify the typical error associated with MIPAS retrieved results. They may give me a feel for the seasonal dependence of certain errors and where in the vertical profiles these errors may occur, but I struggle to believe the errors have been "quantified" in any meaningful way.

I think the main take home message of the paper is that to accurately retrieve information from the vertical profiles of limb emissions a two-dimensional scheme is required. This is well known. The paper is a report of some work but it's use for the interpretation of MIPAS data is not clear. I would really like the authors to improve their discussion related to how their results guide the reader to better interpret artefacts within the standard MIPAS data products.

Comments and concerns

(page 2-line 7) it is stated that the MIPAS observations are exploited. This work is entirely simulation so I don't see any exploitation of the MIPAS data.

(page 2-line 29) I think the authors should spend some more time justifying the statement that a 1.4 degree model is highly resolved. How does this resolution relate to MIPAS sampling resolution? The paper should do a better job of demonstrating the forward model of the radiance measurements is sufficient to accurately simulate realistic MIPAS measurements. I think the paper is trying to tell me that MIPAS 1-D retrievals have errors so the forward model must be justified as an accurate representation of MI-

PAS measurements in order for me to interpret the two-dimensional results.

(page 3-line20) Is a four day and only eight orbit set of observations sufficient to quantify the errors in the one-dimensional approach? I believe that some specific errors have been quantified but I need more information to know that "the errors" have been quantified. Once again for this paper to be useful it must tell a complete MIPAS related story.

Summary

I found the paper to be well written but without much value in its current state. The results presented need to be linked more to the MIPAS measurements in order for them to be of value. It is very well known that two-dimensional retrievals do a better job of retrieving two-dimensional structure and without a more comprehensive link to the MIPAS data set this paper simply reiterates this well known fact. If the authors attempt to link their simulated results to the existing MIPAS data sets in a more realistic fashion I will be happy to look at the paper again.

---

## Author Comment (AC1) · 20 Jul 2016

**Reviewer #1**

The manuscript "Errors induced by different approximations in handling horizontal atmospheric inhomogeneities in MIPAS/ENVISAT retrievals" by E. Castelli et al. evaluates several approaches to account for the horizontal atmospheric inhomogeneity. Many limb algorithms assume that the atmospheric parameters and gas concentrations do not vary horizontally along the line of sight. However, this assumption can produce significant errors in the retrieved profiles, especially over latitudes and seasons where large horizontal gradients in atmospheric parameters are observed. In this study authors test several approaches to account for the horizontal inhomogeneity and provide error estimates for each method. This study fits to the scope of the problems considered in AMT. The paper is well organized and written for most of the part. The methods and results are fully explained. The manuscript is recommended for the publication after some minor corrections.

The authors gratefully acknowledge the reviewer for the time spent in reading the paper and for his/her useful suggestions. The authors' reply to each comment is reported in blue below the reviewer's comment.

Specific comments: Throughout the manuscript authors use many acronyms, but not all of them are explained. Please, spell out all acronyms where they are used for the first time in the text. page 2, l.21-23, Please, spell out acronyms 'GMTR', 'MORSE', 'RET2D' and 'RCP'. You use 'GMTR' several times later in the text. Figure 2, Please, spell out 'ORM'.

Done. RET2D, however is defined as "2D retrieval code" but the acronym is never spelled out.

Page 4, Section 2.2, lines 9-15: I found that this part of the section is not clearly written. Did you set the pressure profile to be the same for all latitudes in order to reduce a natural noise and to better isolate an error due to the 1D assumption? Please, explain that in the text.

This is correct. Since the error due to horizontal pressure gradients is very small compared to that owing to the other horizontal gradients (see also reply to related comment of reviewer #2), we set the pressure profile to be horizontally constant to better isolate the error due to the 1D assumption on T and VMR. In the revised version of the manuscript we added a comment on this regard. We added: "For this reason we set the pressure profiles to be latitudinally constant to better isolate the error due to the 1D assumption on the other targets." after the sentence " The vertical distributions of pressure and the other interfering.." in page 4 line 12 of the original manuscript.

Page 7, lines 5-14. I feel this part needs some revisions. For example, the text says "H2O RMSEs can be reduced down to 3 % in the 0.1-70 hPa region and to 10 % in the 70-200 hPa region by modeling gradients". By modeling gradients in T or T+VMR? "The error on O3 reduces to a few percent below 40 hPa and above 0.2 hPa (see Fig. 6)." Why did the O3 error reduce? Did it reduce as a result of modeling temperature gradients or VMR gradients? "This error can be greatly reduced when atmospheric variability is taken into account (Fig. 4 and Fig. 7)". Did you mean the 1D + gradients approach here or the full 2D retrieval?

We changed the sentence on water vapor: "H2O RMSEs can be reduced down to 3 % in the 0.1-70 hPa region and to 10 % in the 70-200 hPa region by modelling gradients" with "H2O RMSEs can be reduced down to 3 % in the 0.1-70 hPa region and to 10 % in the 70-200 hPa region by modeling temperature and VMR gradients".
In case of O3 we added "When both the VMR and Temperature gradients are applied, the error on O3 ..."
For CFCs-11 the sentence "This error can be greatly reduced when atmospheric variability is taken into account (Fig. 4 and Fig. 7)" applies to all the cases analyzed in this work.
Thus we changed the sentence in "This error can be greatly reduced when atmospheric variability

is taken into account (Fig. 4 and Fig. 7) using any of the methods described in this study".

Figure1. Please, add a title on each panel of Fig. 1, for example "Temperature". You have labels on color scales, but they are too small to see.

OK done, we also corrected a typo in the caption and in the text, we removed "H2O" that is not shown in the figure.

Figures 3, 4, 6 and 7: Please, add labels for each color lines. You have these labels on Figs.2 and 5.

OK done.

Figure 5. I would recommend to spell out 'RMSE' here for readers convenience, even though you did that in the text.

OK, in the revised manuscript we spelled out RMSE also in the caption.

---

## Author Comment (AC2) · 20 Jul 2016

**Reviewer #2**

This is a nice paper that clearly describes a solid study of the impact of horizontal gradients on various approaches to retrievals from the MIPAS instrument. In principal I am happy to see this manuscript proceed to AMT, however, I only have one concern that I'd like to understand beforehand (plus some minor suggestions/comments for the authors to consider). The standard of English is excellent and the figures are very clear.

We gratefully acknowledge the reviewer for the constructive comments and suggestions that help in improving the quality of the paper.

The replies to the reviewer's comments are reported in blue below each comment.

My main concern surrounds the discussion on page 4, lines 10-15. As I understand it, the authors have taken model temperature and pressure profiles on a fixed height grid, and forced the pressures to be horizontally homogeneous while retaining the model horizontal temperature variability. If the altitude grid indeed remains fixed, then surely the model atmosphere that results is not in hydrostatic balance. As such, this becomes an unphysical simulation from which it is arguably hard to draw meaningful "real world" insights. Furthermore, are the lines being observed not subject to significant pressure broadening, making pressure the dominant contributor to the radiance signals? As such, I would have thought that horizontal gradients in pressure are arguably the most important thing to study the impact of (though one would probably ultimately quantify it in terms of impact to the temperature/composition profiles as interpolated to a fixed pressure grid, being the product most widely used in the community). While I understand the authors desire to "focus the analysis on the error caused solely by the approximations in modeling the horizontal variability of temperature and target gas", surely, if the pressure gradients are indeed the dominant term, they should have been included in the analysis. I'd like the authors to address this point, and consider revising their approach, or at least undertaking a separate quantification of the impact of pressure gradients (but again, the unphysical nature of their atmosphere would limit the usefulness of the result).
Perhaps I have misunderstood the description in the manuscript, in which case, greater clarity is required.

The reference atmosphere used for the analysis was made using temperature and pressure profiles from the model on a fixed altitude grid, and forcing the pressures to be horizontally homogeneous while retaining the model horizontal temperature variability (as correctly stated by the reviewer). Then we retrieve both temperature and pressure (plus VMR when required) on a fixed altitude grid, retaining the horizontal variability of either T or VMR as in the model if 1D + gradient retrievals are performed. Since the retrievals are all performed with the GMTR code and into the GMTR code the pressure is left free to vary during the p,T retrieval procedure without imposing hydrostatic equilibrium, we obtain T and P on a fixed vertical grid. The retrieved pressure is then used as vertical coordinate for the retrieved Temperature when comparing the profiles with the reference ones. Then, as correctly stated by the reviewer, the final impact is evaluated on temperature/composition profiles on a fixed pressure grid.
The reviewer's main concern is that such an approach, where the pressure is considered as horizontally homogeneous and no horizontal pressure gradient is accounted for, is not enough realistic to reproduce real measurement conditions. At the beginning of this work, to test the capability of our scheme to correctly reproduce real MIPAS L2 behavior as part of ESA-ESRIN Contract no. 21719/08/I-OL, we compared the 1D AX-DX differences obtained with our approach with the ones calculated with ESA level 2 products (real data) for the same months in several years. Apart from differences due to day/night conditions (not modeled in our study) in general the ESA 1D AX-DX differences and our 1D results matches well in both shapes and amplitude. These comparisons are reported in the Technical note "Investigations on horizontal inhomogeneities issue: Outcome of WP 9000".
Further tests were performed comparing the results we get with the 1D and 1D+T gradient approach with the ones reported in Kiefer et al. 2010. As also stated in the manuscript, we compared our results for T, HNO3 and CFC-11 in terms of AX-DX differences with the one in

Kiefer et al., 2010 and find similar results. All these findings suggest that 1) the scenario used in our test is realistic enough to draw meaningful conclusions and that 2) the pressure gradients have a second order effect with respect to temperature and VMR gradients.

In addition, following the reviewer's suggestion, we performed a test using the reference atmosphere of 21th March in which both pressure and temperature are not forced to be horizontally homogeneous. Then we performed a 1D and 1D+T gradient retrieval and calculated the differences with respect to the reference (not homogeneous in pressure) Temperature field. These results are finally compared to the ones used for the analysis reported into the manuscript (obtained with the horizontal homogeneous pressure field). As we can see from Figure R1 reported below, we can hardly discriminate between the case where the pressure is horizontally homogeneous (left column maps) or not (right column maps). In both cases the temperature horizontal gradient is by far the major contributor to the 1D error. This finding is in agreement with what stated in Kiefer et al., 2010: "Furthermore, the facts that in the 1-D temperature retrievals there is already a clear effect, and that species retrievals from mid-IR emission spectra strongly depend on temperature, suggest to refine this hypothesis: the major part of the ascending/descending differences of 1-D retrieval results is caused by the neglect of the horizontal temperature inhomogeneities in the retrieval algorithm."

[Figure]

**Figure R1: Left column: Temperature field retrieved with 1D code minus reference (top) and 1D+gradT minus reference (bottom), using in the reference atmosphere a horizontally homogeneous pressure field. Right column: Temperature field retrieved with 1D code minus reference (top) and 1D+gradT minus reference (bottom) using a reference atmosphere with horizontally inhomogeneous pressure field.**

In order to quantify the impact of pressure gradients, as suggested by the reviewer, we built a reference atmosphere for the 21 of Mach where both temperature and composition are horizontally

homogeneous while only the pressure profiles varies with latitude. Then we performed a 1D retrieval. The impact on temperature field is evaluated for this analysis comparing the results with the ones obtained using the 2D retrieval approach. The results are reported in Figure R2. As we can see from this figure, the effect of pressure gradients on 1D retrieval is negligible and it is of the same order of the smoothing error component.

[Figure]

**Figure R2: RMSE error on temperature due to pressure gradients on 1D retrievals (blue line) and on 2D (yellow, smoothing error). The grey areas represent the average ORM random error.**

Finally, in the frame of MIPAS QWG (Quality Working Group) activities, we performed some tests with real data using a 1D+gradient approach. In these tests we also accounted for the horizontal pressure gradients. The results show that accounting or not for these gradients produced negligible differences on AX-DX differences. (see slide 15/49 of https://earth.esa.int/documents/700255/2551278/2.9_Ridolfi_and_Sgheri_testing_orm_v8.pdf).

According to reviewer's suggestions we revised our paper as follows:

1) We now assess the validity of our assumptions in reproducing real measurements behaviour, by comparing the AX-DX differences from simulated data with the ones from ESA IPF V6.0 Level2 MIPAS data computed by M. Kiefer in the frame of the MIPAS QWG activities. We compare those differences for the corresponding month in the years from 2005 to 2010 to our results by overplotting them in figures 2-3-4. In the text we comment about the good agreement between those data by adding at pag.6 line 3 of the original manuscript "For comparison purposes, in the same figures we show the values of AX-DX differences calculated from ESA IPF V6.0 level2 MIPAS data of December 2005 to 2010. Simulated 1D retrievals and real measurements show a very similar behaviour for most of the target species in the altitude range where ESA AX-DX differences are available, despite of the fact that different years are used. The amplitude of the 1D AX-DX differences is comparable to that of real data, confirming the fact that the simulated observations used in our tests are suitable for reproducing the behaviour of real MIPAS measurements." Accordingly in the caption of Figure 2 we added: "in green the differences from ESA IPF V6.0 Level2 MIPAS data in December form 2005 to 2010". M.Kiefer how provided the ESA V6.0 AX-DX differences has been added to the list of authors.

2) At page 4 line 12 of the original manuscript (see also the reply to reviewer#1's comment) we included a sentence about the impact of the missing horizontal pressure variability model in the 1D retrievals. The sentence is: "Further tests, on simulated and real data (private communication), demonstrate that the pressure horizontal gradient has an almost negligible effect on 1D retrievals. For this reason in this study we set the pressure profiles to be latitudinally constant to better isolate the error due to the 1D assumption on the other targets."

==== Minor comments
—- Page 3
Lines 27/28: I don't understand this sentence. By "average atmosphere in a given latitude band" do you mean a zonal mean, i.e., an average over all longitudes? If so, hasn't longitude, by definition, been ruled out. Do you mean averages over a longitude range should not depend on the choice of

range (clearly not an appropriate assumption geophysically). Please clarify.

This sentence refers to the work performed by Kiefer et al., 2010 using real measurements. In their paper the authors use a set of MIPAS measurements covering 10days/one month to calculate AX and DX averages in each latitude band. Given the large size of the considered dataset and considering the MIPAS sampling rate, it turns-out that in a given latitude belt the different longitudes are evenly sampled in the AX and the DX parts of the satellite orbits. For this reason the deviations from zero of the AX-DX differences cannot be ascribed to the different sampling longitudinal intervals in the AX and DX parts of the orbits. In this sense "the average atmosphere in a given latitude band should not depend on longitude therefore, with a perfect retrieval scheme, the zonal averages of retrieved parameters should be equal when computed from measurements in the ascending or descending parts of the orbit".

Since in the model the AX and DX atmosphere could be different, in our one-orbit tests we used a AX-DX symmetric atmosphere. If we would have used a non-symmetric atmosphere, differences due to different longitudinal positions of the profiles in AX and DX part of the orbit could influence the AX-DX calculation.

In order to clarify this point, in the revised version of the paper, we modified the sentence "The results of Kiefer et al (2010) are based on the assumption that the average atmosphere in a given latitude band should not depend on longitude, therefore with a perfect retrieval scheme the zonal averages of retrieved parameters should be equal when computed from measurements in the ascending or descending parts of the orbit." in this way:

"The results of Kiefer et al. (2010) are based on the assumption that the average atmosphere in a given latitude band should not depend on longitude. Actually those averages are computed using several days of measurements allowing for an almost equal longitudinal distribution of ascending and descending profiles. Therefore with a perfect retrieval scheme (and a constant in time atmosphere) the zonal averages of retrieved parameters should be equal when computed from measurements in the ascending or descending parts of the orbit."

—- Page 4
Line 5: Please be more quantitative; what is the spatial resolution used in the FM?

The spatial resolution of the FM is the one of the EMAC atmospheric model: 1.4 degrees. We included this information in the revised version of the manuscript: we modified page 4 line 5 "the discretization of the atmosphere is sufficiently fine .." into "the discretization of the atmosphere is sufficiently fine (1.4 °)".

Lines 10-15: Please see discussion above.

Please see answer above.

Lines 17-19: First, please explain why you needed to add any noise at all? Why not simply do a noise free simulation. Also, please state whether, in addition to adding 1/40th of the expected noise, you also quote that 1/40th value as the radiance precision in the retrieval calculation (S_y in the Rodgers formulation), or does the retrieval still believe that the noise is at the 100% level?

We use 1/40th of the nominal noise specification both to add a synthetic pseudo-random error to the simulated measurements and to define their error covariance matrix Sy in the retrieval. Due to the different discretizations of the atmosphere in the FM that simulates synthetic measurements and in the FM internal to the inversion algorithm, we find that convergence of the retrieval is much more difficult if no noise is used. On the other hand the noise error must be significantly reduced with respect to the nominal case in order to make the horizontal model approximation the main source of error as already performed in in Carlotti et al., 2013 for the evaluation of the position error on MIPAS 1D retrievals. This sentence has been added into the revised manuscript page 4 line 27

—- Page 5
Line 20 (and 22). Your "diff" has been typeset in math mode, you'll want to typeset it in text mode

(e.g., \text{diff}, using the amsmath.sty package).

Ok, done thanks.

—- Page 6
Line 26: I think this would be better "Polar and mid-latitude winter conditions" if that's what you mean. As it is it could mean "Polar winter" and "mid-latitude all seasons".

It is "Polar winter" and "mid-latitudes" all seasons as described in section 2.4 page 5 lines 28-29.

—- Page 7
Line 2: As for page 6, line 26. Possibly elsewhere that I didn't catch also.

See answer to comment above.

—- Figures
Figures 2 and on: I would much prefer to see temperature errors quantified in K than in %. K are much more accessible to the general reader.

Percentage values are used only in Figure 2, in Figure 5 and onward we used K. According to the reviewer's suggestion, in the revised version of the paper we changed Figure 2 using absolute values in K instead of percentage values for AX-DX differences.

---

## Author Comment (AC3) · 20 Jul 2016

**Reviewer #3**

Review of "Errors induced by different approximations in handling atmospheric inhomogeneities in MIPAS/ENVISAT retrievals" by Castelli et al.

This paper provides simulated results that describe the need for a two-dimensional retrieval approach to infer vertical information from MIPAS limb emission measurements.
The authors simulate measurements and perform a variety of different retrievals in an attempt to quantify the errors induced by an assumption of horizontal homogeneity within the retrieval scheme. The simulations are by no means comprehensive but the reader is left with the impression that any retrieval scheme that uses two-dimensional information is less likely to produce artefacts related to atmospheric structure along the satellite track. This is of course by no means a surprising result. Although this paper provides no real information it is another work that supports the hypotheses that in order to accurately retrieve atmospheric composition information from limb measurements, a set of dense observations and a multi-dimensional retrieval are required.

The authors thank the reviewer for carefully reading the manuscript and for the useful comments. The replies to the reviewer's comments are reported in blue below each comment.

Major Comments:
I found the paper to be well written and the information that was presented was done so in an organized and easily understood manner.

It is very unclear to me what I am supposed to have learned from reading this paper. The limited number of simulations performed do not allow me to quantify the typical error associated with MIPAS retrieved results. They may give me a feel for the seasonal dependence of certain errors and where in the vertical profiles these errors may occur, but I struggle to believe the errors have been "quantified" in any meaningful way. I think the main take home message of the paper is that to accurately retrieve information from the vertical profiles of limb emissions a two-dimensional scheme is required. This is well known. The paper is a report of some work but it's use for the interpretation of MIPAS data is not clear. I would really like the authors to improve their discussion related to how their results guide the reader to better interpret artefacts within the standard MIPAS data products.

The reviewer's concerns are:
**1) Limited number of simulations used to quantify the errors:**
In the standard MIPAS systematic error analysis, the error due to to horizontal inhomogeneities is evaluated by using a temperature gradient of +/-1K/100 km (see Dudhia et al., 2002 and http://eodg.atm.ox.ac.uk/MIPAS/err/), while no error due to VMR horizontal inhomogeneity is considered. In Carlotti et al. 2013, the authors used a more sophisticated horizontal variability model with respect to the +/-1K/100 km to evaluate the so called "position" error on 1D retrievals. In this case the authors used a single atmosphere retrieved from real MIPAS measurements acquired during one orbit. Our analysis is based on simulations of 4 days of measurements in four different seasons to assess the errors due to neglecting the horizontal variability or to using a simplified approach to model it. This is a clear improvement with respect to the existing assessments.
2) **Relations between simulated retrieval results and MIPAS standard products not clear:**
As reported also in a reply to reviewer#2's comment, in the frame of the ESA-ESRIN Contract no. 21719/08/I-OL, we compared the 1D AX-DX differences obtained with our synthetic spectra with the ones calculated with ESA products when analysing real measurements for the same months in several years from 2005 to 2010. Apart from the differences owing to the day/night variations of the atmosphere, that are intentionally excluded from our simulations, the AX-DX differences derived form the ESA v6 1D products match pretty well both in shape and amplitude. These comparisons are reported in the Technical note "Investigations on horizontal inhomogeneities issue: Outcome of WP 9000". An example of the comparison for temperature is reported in Figure R1 below. Considering all the limitations of the simulations and the fact that in case of real measurements

part of the differences, especially at high altitudes are due to day-night variability of the atmosphere (see e.g. solar tides and changes due to photochemistry), the overall agreement is good.

[Figure]

**Figure R1: Temperature AX-DX differences from simulations for the four days considered in our study and from ESA V6 retrievals for months in corresponding season (e.g. December January February for 21 December) and years from 2005 to 2010.**

In order to improve the paper, as suggested by the reviewer, this information is now included in the

revised version of the manuscript. We compare the AX-DX differences calculated with our simulated observations for the 21 December in the -45/-65 latitude band with those extracted from ESA Level 2 V6 monthly means for the corresponding month from 2005 to 2010, for each target parameter.

The detailed report of the changes we introduced in the manuscript due to these considerations are listed below, in the replies to comments labeled as "page 3-line20" and "Summary".

These comparisons demonstrate that the synthetic measurements used in our tests and the analysis performed with those measurements can be considered representative to assess the error in MIPAS Level 2 products, due to different approximations in modeling the horizontal variability of the atmosphere. Thus we conclude that the results of our study can really be used for the interpretation of MIPAS retrieved profiles.

Following these considerations, we conclude that the results we show in the paper are 1) a clear improvement with respect to what was done so far to assess the errors due to neglecting horizontal variability, 2) representative when compared to real MIPAS data and thus can be used for the interpretation of real retrievals and for the evaluation of the different strategies used to cope with horizontal inhomogeneities.

For these reasons we think the work presented in the paper provides real information to the reader.

Comments and concerns

(page 2-line 7) it is stated that the MIPAS observations are exploited. This work is entirely simulation so I don't see any exploitation of the MIPAS data.

We agree with the reviewer. Even if in the revised version of the paper we also add AX-DX differences calculated from ESA level 2 MIPAS data, the error estimation is based on synthetic observations. Thus in the revised version of the manuscript we changed the text accordingly: we replace "the measurements of" with "synthetic observations simulated for" in page 2 line 7 of the original manuscript.

(page 2-line 29) I think the authors should spend some more time justifying the statement that a 1.4 degree model is highly resolved. How does this resolution relate to MIPAS sampling resolution? The paper should do a better job of demonstrating the forward model of the radiance measurements is sufficient to accurately simulate realistic MIPAS measurements. I think the paper is trying to tell me that MIPAS 1-D retrievals have errors so the forward model must be justified as an accurate representation of MIPAS measurements in order for me to interpret the two-dimensional results.

Regarding the horizontal resolution: the horizontal sampling of MIPAS nominal limb-scans is about 4 degrees, approximately equal to the horizontal resolution of the measurements. In this sense a model atmosphere with 1.4 degrees resolution is "highly resolved with respect to the MIPAS sampling". Following the reviewer's suggestion we included this information in the revised text: "(1.4° in latitude, much finer than the MIPAS horizontal sampling of about 4°)".

Regarding the forward model: We recall here that the 2D Forward Model (FM) used in the paper is the one internal to the 2D GMTR code. The capability of this FM to correctly model MIPAS measurements accounting for the horizontal variability was clearly demonstrated in Kiefer et al., 2010. In this paper the authors show that the AX-DX differences calculated from 2D GMTR retrieved profiles are very similar to those calculated from ECMWF profiles. The AX-DX differences owing to the 1D retrievals are always greater. This is a solid demonstration, based on real data, of the capability of the 2D FM to correctly reproduce MIPAS measurements and to model the horizontal variability. As suggested by the reviewer we included this information in the revised version of the manuscript (section 2.2 "Synthetic observations", page 4 line 4 of the original manuscript) : "In Kiefer et al., 2010 the authors demonstrate the capability of the GMTR code to correctly model the features of MIPAS measurements due to the horizontal variability of the atmosphere. AX-DX differences calculated from profiles retrieved with the GMTR code are very similar to those calculated with the corresponding ECMWF data. In contrast, AX-DX differences calculated from profiles derived with 1D codes show features (not present in ECMWF data), hence

related to an incorrect modelling of atmospheric horizontal variability."

(page 3-line20) Is a four day and only eight orbit set of observations sufficient to quantify the errors in the one-dimensional approach? I believe that some specific errors have been quantified but I need more information to know that "the errors" have been quantified. Once again for this paper to be useful it must tell a complete MIPAS related story.

As above, the main reviewer's concern here is that the set of simulated observations used in the paper is not enough representative of the natural horizontal variability encountered by real MIPAS measurements to correctly quantify the errors of the 1D approach. In the revised paper we address this point by comparing the effect of horizontal variability on 1D retrievals performed with both real and our simulated data. As a quantifier for the error implied by the 1D approximation we use the AX-DX differences, that represent also the first experimental evidence of the effect of the horizontal homogeneity assumption on MIPAS-ESA 1D retrievals. As mentioned above, we compared the 1D AX-DX differences obtained with our synthetic spectra with those calculated from ESA products, for the same months of several years from 2005 to 2010. The good agreement between the observed and simulated AX-DX differences is both in shape and amplitude. This test proves that the simulated set of observations is suitable to pursue the objective of the paper, that is the assessment of the error implied by different approximations in modelling the horizontal variability of the atmosphere.

 Owing to these considerations, we modified paper as follows:
1) In Fig.s 2, 3 and 4 we included also the curves related to the the ESA v6 AX-DX differences for the month of December, in the years 2005, 2006, 2007, 2008, 2009 and 2010.
2) We included a comment regarding the good agreement between observed and simulated AX-DX differences. The comment is in pag.6 line 6 of the original manuscript " For comparison purposes, we report in the same figures the values of AX-DX differences calculated from the ESA IPF V6.0 level2 MIPAS data of December 2005 to 2010. Simulated 1D retrievals and real measurements show a very similar behaviour for most of the target species in the altitude range where ESA AX-DX differences are available, despite the fact that different years are used. The amplitude of the 1D AX-DX differences is comparable to that of real data, confirming the fact that the simulated observations used in our tests are suitable for reproducing the behaviour of real MIPAS measurements." Accordingly in the caption of Figure 2 we added: "in green the differences from ESA IPF V6.0 level2 MIPAS data in December 2005 to 2010". M. Kiefer who provided these differences has been added as an author and removed from acknowledgements.

If needed a supplement reporting analogous differences calculated for the other four days can be added (e.g. as Figure R1 in this document).

Summary
I found the paper to be well written but without much value in its current state. The results presented need to be linked more to the MIPAS measurements in order for them to be of value. It is very well known that two-dimensional retrievals do a better job of retrieving two-dimensional structure and without a more comprehensive link to the MIPAS data set this paper simply reiterates this well known fact. If the authors attempt to link their simulated results to the existing MIPAS data sets in a more realistic fashion I will be happy to look at the paper again.

We have already replied above to the reviewer why we think that the paper contains valuable information for the reader.  In addition, as stated above, in the revised version of the paper we address the reviewer's concern and link our results, obtained with simulated spectra, with those obtained with real MIPAS measurements. This is done by including in plots and discussion the results from ESA V6 level 2 data. In particular in figures 2-3-4 we included the AX-DX differences calculated from ESA v6 data for December of the years from 2005 to 2010. The figures show that the real and simulated AX-DX differences obtained with the horizontal homogeneity assumption agree quite well both in amplitude and shape.  Real AX-DX differences observed in ESA Level 2 v6

products in the days corresponding to the four seasons, and in all latitude bands can be given provided as a supplement, if required.